# Evidence for impacts on surface-level air quality in the Northeastern U.S. from long-distance transport of smoke from North American fires during LISTOS 2018

Haley M. Rogers[1], Jenna C. Ditto[1], Drew R. Gentner[1,2]

[1] Department of Chemical and Environmental Engineering, Yale University, New Haven, CT, 06511, USA

[2] SEARCH (Solutions for Energy, Air, Climate and Health) Center, Yale University, New Haven, CT, USA.

*Correspondence to*: Drew R. Gentner (drew.gentner@yale.edu)

**Abstract.** Biomass burning is a large source of uncontrolled air pollutants, including particulate matter (i.e. $PM_{2.5}$), black carbon (BC), volatile organic compounds (VOCs), and carbon monoxide (CO), which have significant effects on air quality, human health, and climate. Measurements of $PM_{2.5}$, BC, and CO made at the Yale Coastal Field Station in Guilford, CT and five other sites in the metropolitan New York City (NYC) area indicate long-distance transport of pollutants from wildfires and other biomass burning to surface-level sites in the region. Here, we examine two such events occurring on August 16th-17th and 27th-29th, 2018. In addition to regionally-consistent enhancements in the surface concentrations of gases and particulates associated with biomass burning, satellite imagery confirms the presence of smoke plumes in the NYC-Connecticut region during these events. Backward-trajectory modeling indicates that air masses arriving at surface-level sites in coastal Connecticut on August 16th-17th passed over the west coast of Canada, near multiple large wildfires. In contrast, air parcels arriving on August 27th-29th passed over active fires in the southeastern United States. The results of this study demonstrate that biomass burning events throughout the U.S. and Canada (at times more than 4000 km away), which are increasing in frequency, impact surface-level air quality beyond regional scales, including in NYC and the northeastern U.S.

Keywords: LISTOS (Long Island Sound Tropospheric Ozone Study), biomass burning, wildfires, urban air quality

## 1 Introduction

Biomass burning, which occurs on a large scale during wildfires and some controlled burns, is a major source of air pollutants that impact air quality, human health, and climate (Lewis et al., 2008; Liu et al., 2015; Reid et al., 2016; Urbanski et al., 2008). During these events, gases such as carbon monoxide (CO), carbon dioxide ($CO_2$), methane ($CH_4$), nitrous oxide ($N_2O$), nitrogen oxides ($NO_x$), and gas-phase organic compounds (including volatile organic compounds (VOCs)) are directly released into the atmosphere (Akagi et al., 2011; Urbanski et al., 2008; Vicente et al., 2013; Yokelson et al., 2013). Biomass burning produces particulate matter (PM), including black carbon (BC) and other primary organic aerosol (POA) in the $PM_{2.5}$ size range (i.e. particles with a diameter $\leq$ 2.5 μm) (Akagi et al., 2011; Urbanski et al., 2008). Biomass burning is also a source of reactive precursors to the production of secondary compounds, such as ozone ($O_3$) and secondary organic aerosol (SOA) (Urbanski et al., 2008; Ward and Hardy, 1991). The chemical composition of PM resulting from biomass burning depends on many factors, such as the type of fuel and combustion conditions (Calvo et al., 2013). In addition to the environmental impacts of biomass burning emissions, elevated $PM_{2.5}$ concentrations have been associated with respiratory and cardiovascular disease, and higher mortality rates (Brook et al., 2004; Dockery et al., 1993; Reid et al., 2016).

The pollutants emitted from biomass burning events affect not only local air quality, but can be transported over long distances (Barnaba et al., 2011; Burgos et al., 2018; Forster et al., 2001; Martin et al., 2006; Niemi et al., 2005; Stohl et al., 2003). Colarco et al. (2004) used satellite and other remote-sensing tools, combined with backward-trajectory and 3-D models, to confirm the presence of pollution from July 2002 wildfire smoke that originated in Quebec, Canada and was transported and detected at surface-level in Washington, D.C. Similar studies have described the long-range transport of wildfire smoke from Canadian wildfires to Maryland (Dreessen et al., 2016), Siberian wildfires to British Columbia (Cottle et al., 2014), as well as examples in Europe and Asia (Diapouli et al., 2014; Jung et al., 2016). Over the course of this long-distance transport, the gas- and aerosol-phase compounds undergo aging and dilution. Organic gases and aerosols are transformed chemically by photo-oxidation, interaction with atmospheric oxidants, and reaction with other atmospheric compounds (Cubison et al., 2011; Hennigan et al., 2011). While more reactive components will age more quickly, this study focused on tracers which are less likely to react over our transport timescales. For example, BC is primarily removed via particle deposition to the Earth's surface, which is largely dependent on height above ground level. Mixing of aloft plumes from the free troposphere is variable and can range from 1 week to 1

month with altitude, vertical transport conditions, and weather (Jacob, 1999), and $PM_{2.5}$ losses due to physical
processes will follow similar timescales. Losses of these tracers is possible depending on timescales and weather
conditions (e.g. wet deposition) during long-distance transport. However, they are generally long-lived in the free
troposphere and many previous studies have used BC, CO, and/or $PM_{2.5}$ as indicators of long-distance transport of
biomass burning smoke (Burgos et al., 2018; Cottle et al., 2014; Diapouli et al., 2014; Dreessen et al., 2016; Forster
et al., 2001; Martin et al., 2006; Niemi et al., 2005).

The impacts of wildfire smoke, both regionally and at long distances, will become increasingly important in the
coming years, with the number and severity of wildfires predicted to increase with climate change. Barbero et al.
(2015) used 17 global climate models to evaluate the effect of anthropogenic climate change on large-scale wildfires
in the U.S., and found that the likelihood of forest fires will increase across most historically fire-prone regions, likely
due to an earlier onset of summer and extended summer season. Abatzoglou and Williams (2016) similarly found that
wildfires are likely to increase in the coming years due to climate change impacts such as increased temperature and
decreased atmospheric water vapor pressure. As the risks of climate change and its relation to wildfires are realized,
it is increasingly important to understand the environmental and health effects that may be associated, including long-
distance transport.

The NYC metropolitan area (including parts of Connecticut and New Jersey) is home to approximately 20.3 million
people (U.S. Census Bureau, 2017) and has historically struggled with attainment to air quality standards. The
objective of this work is to evaluate the influence of North American biomass burning events on air quality in NYC
and the Northeast U.S. using measurements from the Yale Coastal Field Station (YCFS) in Guilford, Connecticut (on
the Long Island Sound) and other sites in the metropolitan NYC area, combined with satellite imagery and air parcel
backward-trajectory modeling. We focus on observations of two multi-day air pollution events during the month of
August 2018 during the LISTOS (Long Island Sound Tropospheric Ozone Study) 2018 field campaign, both of which
coincided with NYC air quality advisory period for ozone on August 16[th], 28[th], and 29[th] (New York Department of
Environmental Conservation, 2018).

## 2 Materials and Methods

We perform a multi-platform-based analysis to determine whether specific regional air pollution events occurring in coastal Connecticut and the NYC area can be attributed to long-distance transport of emissions from wildfires and other biomass burning. This analysis combines results from pollutant measurements taken at the YCFS and other regional sites, satellite imagery (NOAA Smoke Maps), and the NOAA HYSPLIT backward-trajectory model. Each of these techniques provides some evidence of the long-distance transport of wildfire pollutants, and we combine these methods to evaluate potential sources and transport times.

### 2.1 Yale Coastal Field Station Air Quality Measurements

Ambient surface-level measurements were collected at the YCFS, located on the Long Island Sound in Guilford, CT (41.2583°N, 72.7312°W) using reference instrumentation for PM$_{2.5}$, BC, and CO at 1 hour resolution for PM$_{2.5}$, 1 minute for BC, and 1 second for CO (BC and CO then averaged to 1 hour intervals). An AE33 Aethalometer (Magee Scientific) was used to measure BC; a BAM-1020 (Met One) was used to measure PM$_{2.5}$; and a 48i CO analyzer (Thermo Fisher) was used to measure CO. All instrument flow rates were calibrated, relevant zeroing procedures were performed for the BC and PM$_{2.5}$ measurements, and the CO instrument zero and span concentrations were calibrated (using house-generated zero air and a CO standard from AirGas: ±5% standard 10 ppm CO in nitrogen diluted with AliCat mass flow controllers). We corrected for CO calibration drift when necessary by adjusting the baseline to regional background levels. Inlets for each of these instruments were positioned ~5 m from the water on a small tower 2.5-3 m above the ground, facing south (i.e. towards the Long Island Sound) with direct inflow from the water during southerly onshore winds. Particulate inlets used PM$_{2.5}$ cyclones and metal tubing (BC: copper; PM$_{2.5}$: stainless steel). The CO inlet was constructed of FEP tubing (¼" OD) and PM was removed at the inlet using a PTFE filter (Tisch) and PTFE filter holder. The YCFS is strategically positioned to minimize local urban influence from Connecticut while also being in the NYC metro area. Thus, it serves as a regional background site with less local influence than more urbanized stations.

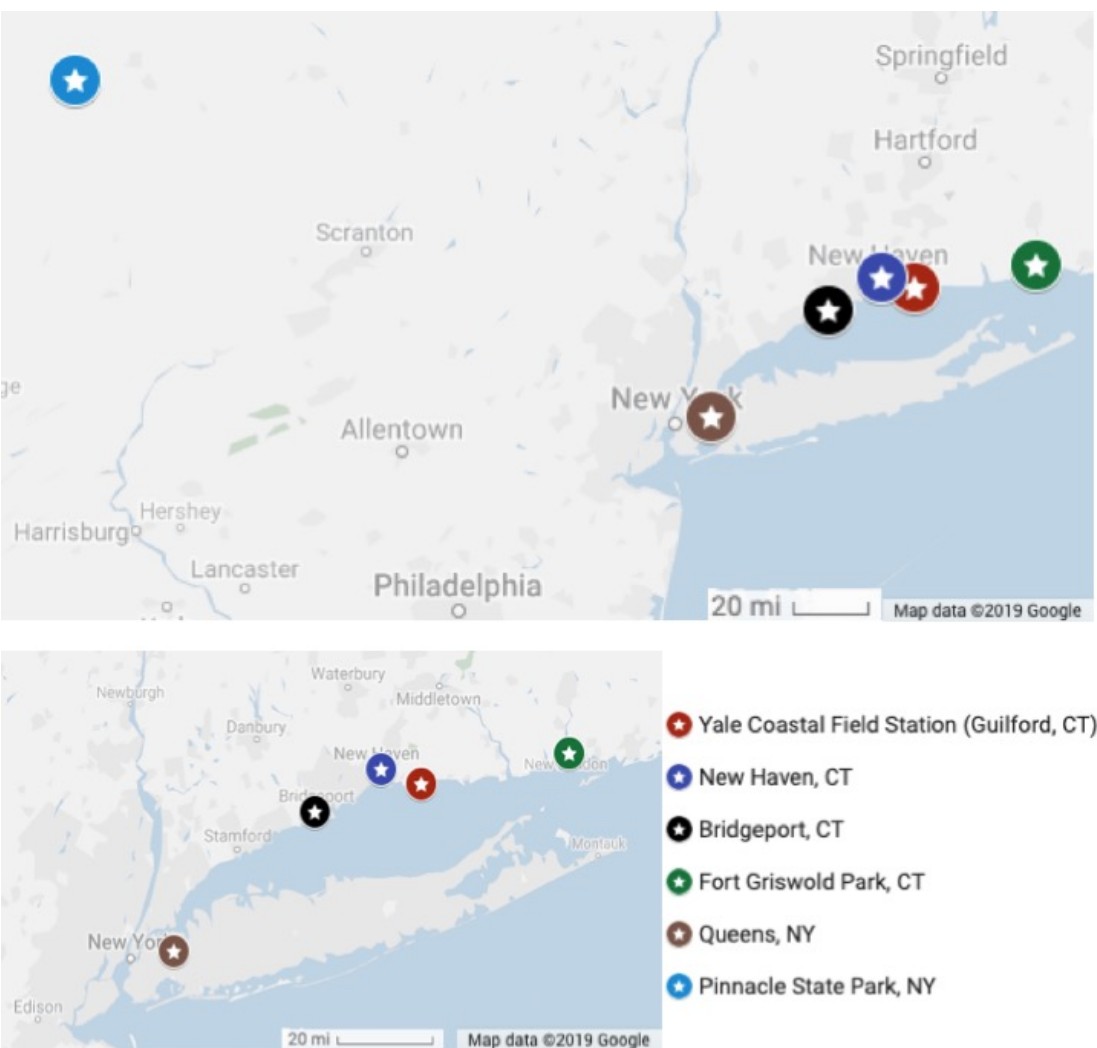

**Figure 1: Location of air quality monitoring sites used for PM$_{2.5}$, BC, and CO measurements. Panel A shows all six sites, while panel B shows a close-up of the five sites directly on the Long Island Sound.**

These YCFS measurements were compared to data from other field sites (for the pollutants available) in the region (Figure 1), including EPA-related sites in New Haven, CT (Site 09-009-0027), Bridgeport, CT (Site 09-001-0010), Fort Griswold Park, CT (Site 09-011-0124), and Queens, NY (Site 36-081-0124), as well as data from the New York Department of Environmental Conservation's rural site in Pinnacle State Park, NY. Sites were selected for regional proximity to the YCFS as well as data availability.

**2.2 Satellite Imagery of Smoke Plumes**

The NOAA Hazard Mapping System (HMS) generated Smoke Maps (NOAA, 2018) once a day based on satellite imagery of the spatial distribution of visible smoke plumes across North America. The data were downloaded from

the NOAA smoke products website and mapped via Google MyMaps. While these maps do not provide vertical
resolution on the distribution of the smoke plumes, they provide information on the horizontal distribution and density
of the smoke plumes in the region.
**2.3 NOAA HYSPLIT Air Parcel Backward-Trajectory Modeling**
The NOAA Hybrid Single Particle Lagrangian Integrated Trajectory (HYSPLIT) online software (Stein et al., 2015)
was used to run backward-trajectory models of air parcels arriving at the YCFS during the two periods of elevated
$PM_{2.5}$, BC, and CO. The HYSPLIT model used archived meteorological data to trace the transport of an air parcel
both vertically and horizontally through the atmosphere. The backward-trajectory model was run using GDAS1.0
meteorological data over a 240 hour (i.e. 10 day) period. While longer backward-trajectories can lead to greater
uncertainty, the general trends remain valuable and 10 days is within a time length commonly studied in past work
that utilized HYSPLIT backward-trajectory modeling (Bertschi and Jaffe, 2005; Córdoba-Jabonero et al., 2018;
Creamean et al., 2013; Huang et al., 2010; Smith et al., 2013). A new backward-trajectory was simulated for air parcels
arriving every 3 hours at the YCFS, at a final elevation of 10 m above surface level. We combined all trajectories
simulated during each event observed at the YCFS and the reported North American fires during the period of interest
into collective maps using ArcGIS.
**3 Results and Discussion**
**3.1 Elevated $PM_{2.5}$, BC, and CO at the Yale Coastal Field Station and Other Regional Sites**
Two main events in August (August 16[th]-17[th] and 27[th]-29[th]) caused regional concentrations of $PM_{2.5}$, BC, and CO to
all significantly increase for approximately two- and three-day periods, respectively. Figure 2 shows the
concentrations measured at the YCFS compared to concentrations measured at nearby sites. The pollution events are
multi-day enhancements that are significantly elevated from typical baseline concentrations with some short-term
variability in the hourly data observed at only a single site, and thus attributed to local emissions. $PM_{2.5}$ concentrations
show strong agreement between different field sites in CT and NY, especially during the two events, confirming that
the concentration enhancements were caused by regional changes and not just local sources. Regional BC
concentrations show general agreement across the sites with BC data, as well as apparent diurnal patterns at the urban
New Haven site, likely from local emissions. However, daily baseline concentrations from the New Haven site are in
good agreement with the YCFS site up the coast. The Pinnacle Park site, 300+ km west in upstate New York, is
affected by the initial arrival of smoke plumes, but BC concentrations decrease sooner than at the YCFS or New
Haven, CT site, which is consistent with the eastward movement of the plumes in the satellite imagery (Figures 3 and
4) and backward-trajectories (Figures 5 and 6). CO concentrations have clear multi-day increases at all three field sites
during the two identified pollution events. The two urban sites, especially Bridgeport, CT, have greater diurnal changes
in CO, potentially caused by local sources (e.g. gasoline-powered motor vehicles), while the YCFS site is generally
less affected by local urban emissions.

Some smaller pollutant enhancements are observed earlier in August (August 6th-7th and August 10th). However, these
events have overall lower concentrations than the two events identified on August 16th-17th and August 27th-29th, and
satellite smoke maps show minimal smoke influence in the NYC region, with the exception of August 6th (Figure S2).
Thus, they were not included in the primary analysis. However, it should be noted that August 5th, 6th, 7th, and 10th
were all days where New York State issued an Air Quality Health Alert, primarily for high ozone (New York
Department of Environmental Conservation, 2018).

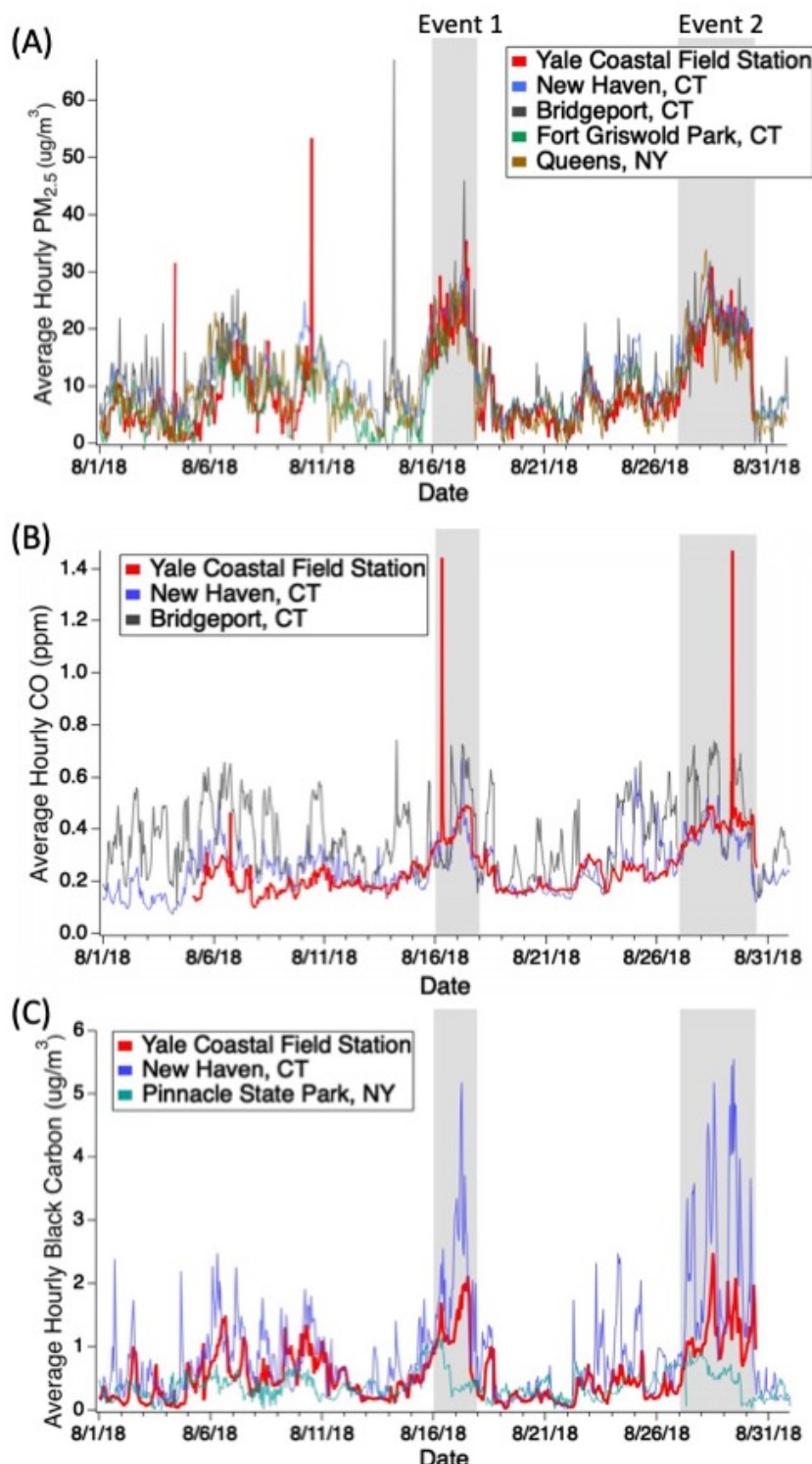

152 **Figure 2: Concentrations of PM$_{2.5}$ (panel A), CO (panel B), and BC (panel C) measured at the YCFS over the month of**
153 **August, 2018. Grey areas represent the two event periods identified as pollution spikes potentially caused by biomass**
154 **burning smoke transport (August 16$^{th}$-17$^{th}$ and August 27$^{th}$-29$^{th}$). These events all show simultaneous increases in PM$_{2.5}$,**
155 **CO, and BC across all field sites, well above baseline concentrations. Meteorological dynamics at Pinnacle State Park, NY**
156 **(300+ km west) appear to be significantly different and lead to different absolute concentrations and earlier event**
157 **dissipation compared to the other sites to the east. Note that the two outlier spikes in CO at the YCFS (panel B) are not**

**ascribed to long-distance transport and are likely due to a hyper-local source near the site (e.g. vehicle, other engine) that**
**caused a brief spike above background levels, evidenced by concurrent $NO_x$ spikes ($NO_x$ is not discussed further in this**
**analysis).**
**3.2 Satellite Imagery: NOAA Smoke Maps**
NOAA Smoke Maps confirm the presence of smoke over the Long Island Sound area during the regional pollution
events with simultaneous enhancements in surface-level concentrations of $PM_{2.5}$, BC, and CO (Figure 2). This satellite
imagery provides evidence that the transport of smoke from biomass burning may have impacted surface-level air
quality during the two pollution events. The daily NOAA Smoke Maps in Figures 3-4 show vertically-integrated
smoke density before, during, and after these events. Figure 3 shows the arrival of an aloft smoke plume with the total
column smoke density peaking at YCFS on August 16[th] and remaining until the 17[th], consistent with the surface-level
pollution event on the 16[th] and 17[th] (Figure 2). The sharp decrease in surface-level concentrations on the 18[th] is
consistent with the departure of the plume in the satellite imagery (Figure 3E). During the second surface-level
pollution event at the end of August, smoke was observed in the region, although less dense than in mid-August.
Figure 4 shows a plume lingering over the NYC and CT region from August 27[th]-29[th] until the morning of the 30th,
which is consistent with surface-level data. No smoke plumes are observed in the area on August 31[st] (Figure 4F),
which is consistent with low surface-level concentrations (Figure 2).

While the satellite imagery lacks vertical distribution data, the presence of smoke in the region during the same periods
when surface-level concentrations increase supports the hypothesis that smoke from aloft was near the surface or
available for transport to the surface, and led to the increase in concentrations of $PM_{2.5}$, BC, and CO at the YCFS and
other regional sites. However, the vertically-integrated column measurements represented by the smoke maps are not
a perfect prediction of surface-level influence, as shown in the days prior to the actual events (i.e. August 14[th]-15[th],
and August 26[th]). On August 14[th]-15[th] leading up to the first event, there are lower levels of smoke aloft over the region
that are visible in the satellite imagery (Figure 3A, 3B). On August 26[th], leading up to the second event, there is again
a low density of smoke over the region (Figure 4A). However, the presence of smoke visible in satellite imagery on
days when the surface measurements do not show an increase in air pollutants is not in conflict with surface-level
results since it may have been exclusively at higher altitudes. On the days prior to the surface-level events when there
is smoke observed aloft in satellite data, it is possible that it had not yet been transported down to the surface sites at
the YCFS and others in the region, which is further explored using vertical-resolved backward-trajectories at higher
time resolution.

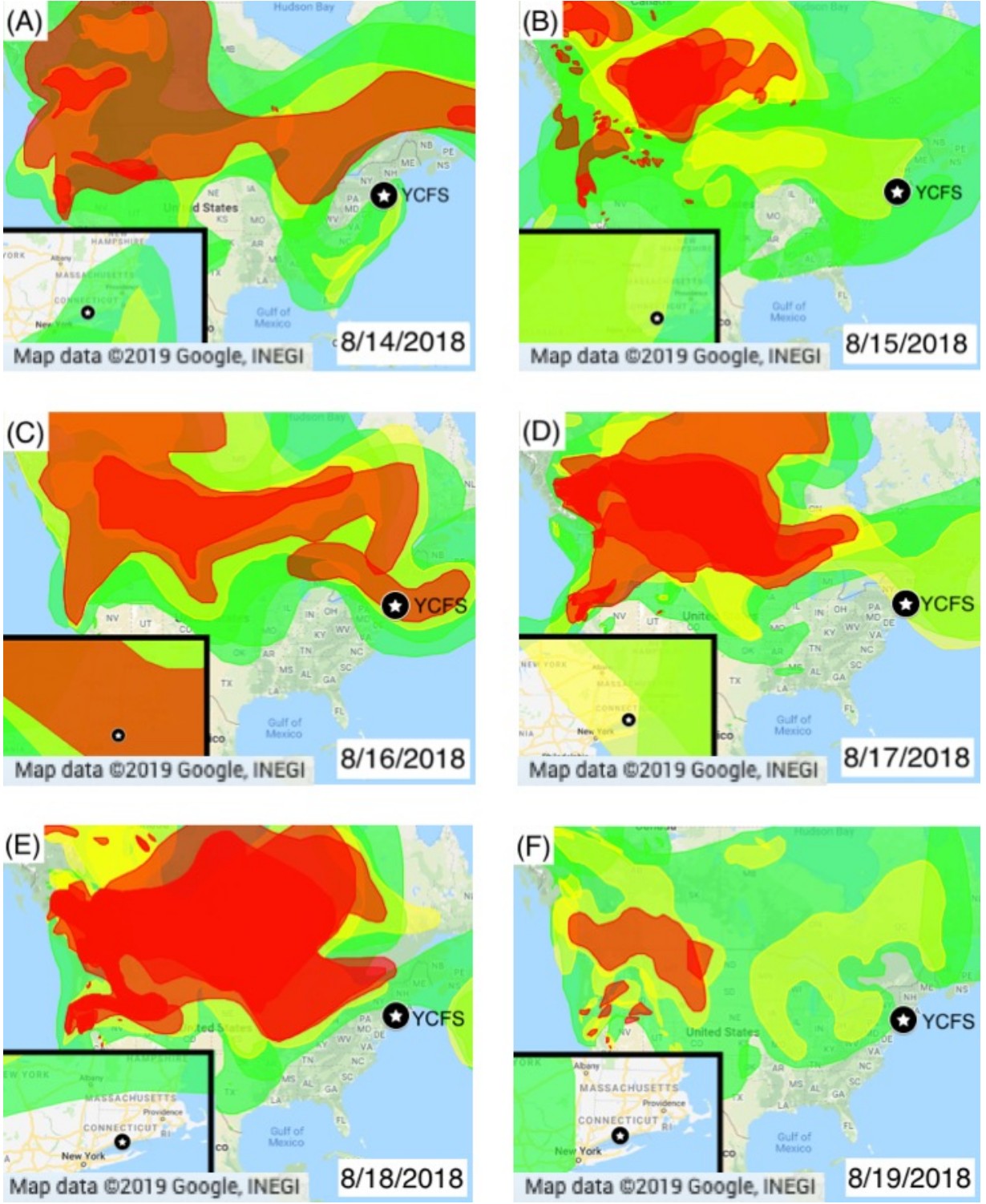


**Figure 3. Smoke Maps (NOAA) based on satellite imagery for total column measurements for August 14th-19th, 2018: before**
**(panels A-B), during (panels C-D), and after (panels E-F) the first surface-level pollution event. The YCFS is indicated by**

a star. A new smoke plume begins to arrive aloft on the 15th before the surface-level pollution event on the 16th and 17th with the aloft total column smoke density peaking on the 16th. The decrease in panels E-F is consistent with the sharp decrease in surface-level concentrations on the 18th. Colors indicate the intensity of the smoke plume, with red being the most dense, yellow intermediate, and green the least dense. Insets provide a magnified view of the YCFS site.

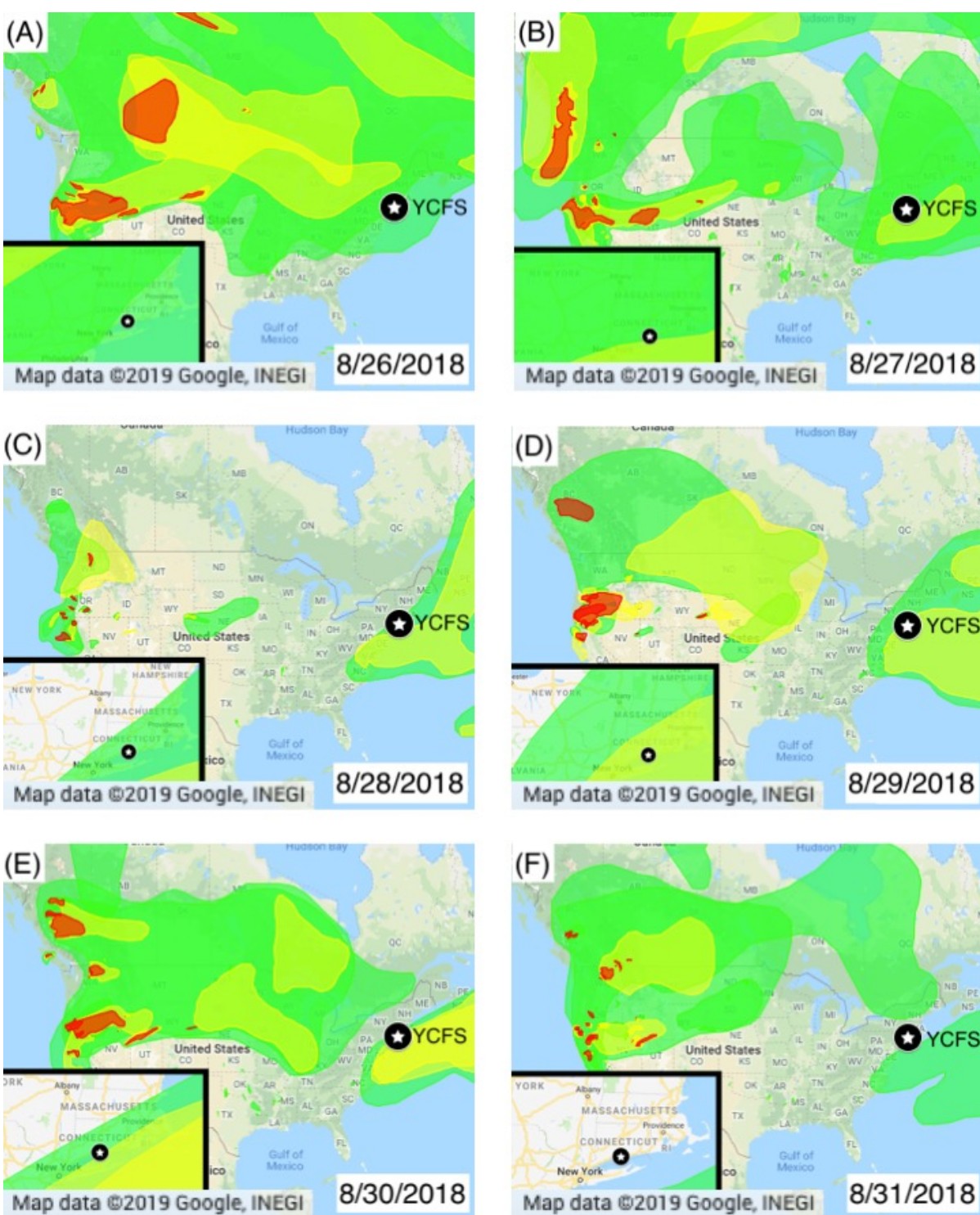

**Figure 4: Smoke Maps (NOAA) based on satellite imagery for total column measurements for August 26th-31st, 2018: before (panel A), during (panels B-E), and after (panel F) the second surface-level pollution event. The YCFS is indicated by a**

**star. The satellite imagery shows a plume lingering over the region during the period of the surface-level event that spanned**
**from August 27th -29th and continued into the morning of the 30th, which is reflected in the satellite imagery. The absence**
**of smoke aloft in panel F is consistent with low surface-level concentrations. Colors indicate the intensity of the smoke**
**plume, with red being the most dense, yellow intermediate, and green the least dense. Insets provide a magnified view of**
**the YCFS site.**
**3.3 HYSPLIT Backward-Trajectory Model Results**
Air parcels originating at surface level in areas with wildfires or controlled burns, or passing aloft over regions where
wildfires were burning, are likely to pick up aerosols and trace gases associated with biomass burning. Here, we use
NOAA HYSPLIT air parcel backward-trajectory models to provide additional information on the horizontally- and
vertically-resolved transport pathways as a function of time of day and potential sources that influenced the observed
surface-level pollution events in the NYC metropolitan area (Figures 5-6).

The backward trajectories for air parcels arriving during the first event (August 16th-17th) show very similar paths
passing over the central coast of western Canada (Figure 5), where NOAA's records of fire locations indicate the
presence of wildfires in this region during the air parcels' transit. On August 16th, the air parcels' backward-trajectory
through Canada and then the northern part of the United States demonstrates that the air parcels passed through an
area with numerous active wildfires and descended from aloft to surface-level in the NYC region on August 16th. On
August 17th, arriving air parcels follow a similar trajectory to the previous day until later in the day (i.e. 15:00 onward)
when air masses did not pass over the North American West Coast within the prior ten days, but stayed in the eastern
half of the United States and Canada in areas without reported fires (Figure 5B). This change in transport pathways
corresponds with a sharp drop in concentrations of pollutants measured at the YCFS observed at the end of August
17th (Figure 2); as wind patterns shifted at the end of August 17th, cleaner air parcels that had not passed through
wildfire regions were transported to the YCFS (Figure 5; greyed out), and thus concentrations of associated pollutants
dropped. These trajectories re-affirm that the spike in pollutant concentrations measured at the YCFS may have
originated from western North American fires for the first event. It is important to note that while most of the
backward-trajectories that passed through active fire regions did not pass within 2,000 meters of the surface (Figure
5B), emissions in forest fire plumes rise due to the heat of combustion and have been shown to commonly reach
heights 2,000-7,000 m above ground level (Colarco et al., 2004; Labonne et al., 2007).

For the air pollution event occurring on August 27[th], 28[th], and 29[th], backward-trajectory modeling shows the majority
of air parcels originated around the Great Lakes region 10 days prior, then circulated in the southeastern U.S. before
arriving to the YCFS. While a few trajectories originate on the West Coast, the majority do not pass through the region
during the 10 day period. However, these air parcels pass over the southeastern U.S. near surface level (~1500 m and
below) where active fires were reported 4-5 days prior to the observed pollution event in the metropolitan NYC region
(Figure 6B). This demonstrates the potential role of biomass burning in the southeastern U.S. for air quality in the
NYC region and northeast U.S. as well. Many of these fires are likely not wildfires, but other biomass burning events
such as intentional crop fires (McCarty et al., 2007). Backward-trajectories for August 27[th] (the start of the second
event) show a similar southeastern circulation pattern as August 28[th] and 29[th] (Figure S3). The last 2 trajectories on
the 29th do not encounter reported fires (Figure 6; greyed out), which is shortly before the dissipation in concentration
at the surface site in the early morning on 8/30 (Figure 2).

237        While satellite smoke maps (Figures 3-4) show the spatial distribution of (vertically-integrated) smoke across

the U.S. during these 2 events, backward trajectories provide more specific evidence that the air parcels observed at
the ground-level YFCS site previously passed over active fires and mixed with biomass burning emissions (e.g. BC,
CO). The fact that smoke is observed via satellite outputs over the YCFS and the NYC metropolitan area during the
same time periods as the ground-level events discussed here, in combination with the fact that that the backward
trajectories passed over reported fires (at altitudes where it was reasonable to expect the rising concentrated smoke
plumes), provides three different pieces of evidence that long-distance transport of biomass burning emissions
impacted air quality in the NYC metropolitan area. In contrast, on many non-event days NOAA Smoke Maps do not
show plumes in the YCFS region and backward-trajectories do not show significant interactions with fire locations
(examples in Figures 3-4, S4-S7).

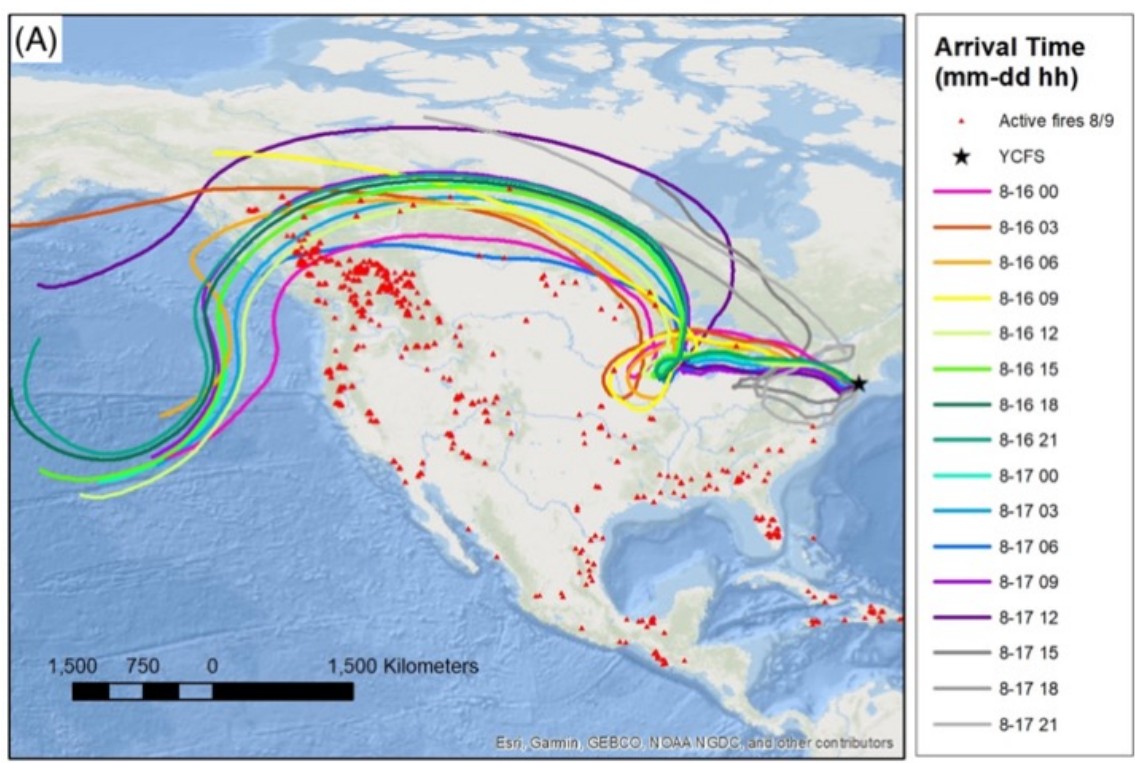

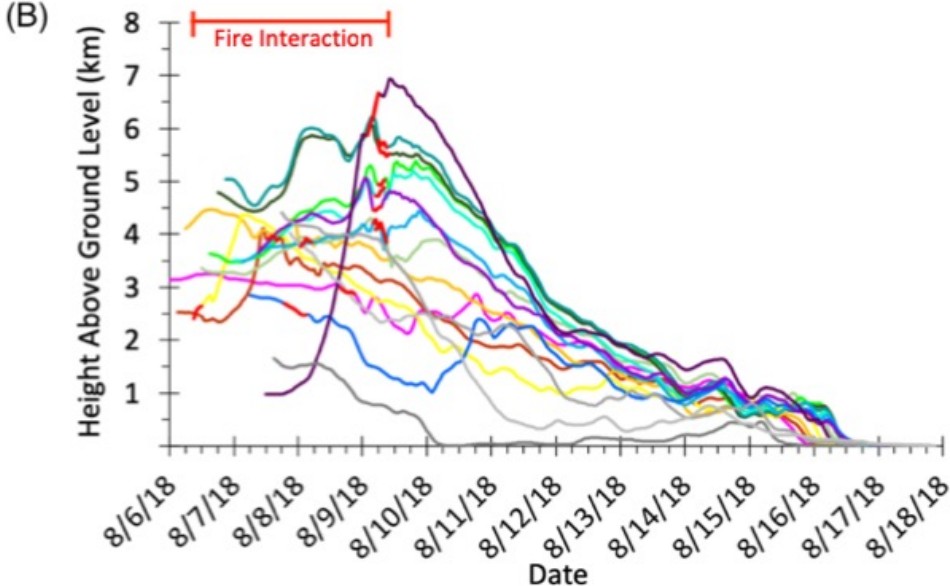


**Figure 5: NOAA HYSPLIT Backward-trajectory model results for air parcels arriving on August 16ᵗʰ and 17ᵗʰ, 2018 to surface-level YCFS site. Each line represents the backward-trajectory for an air parcel arriving every three hours throughout the course of the day. The location of fires on August 9ᵗʰ (when most trajectories intersect the wildfire zone on the West Coast) is depicted with red triangles (from NOAA HMS fire maps). The top map (A) shows the full 10-day trajectory and the bottom figure (B) shows the vertical height of each air parcel along its trajectory as well as the times where it may have intercepted wildfire smoke plumes (highlighted in red on each individual trace, and bracketed above). The last three trajectories on 8/17 have no major fire interaction, and have thus been colored grey.**

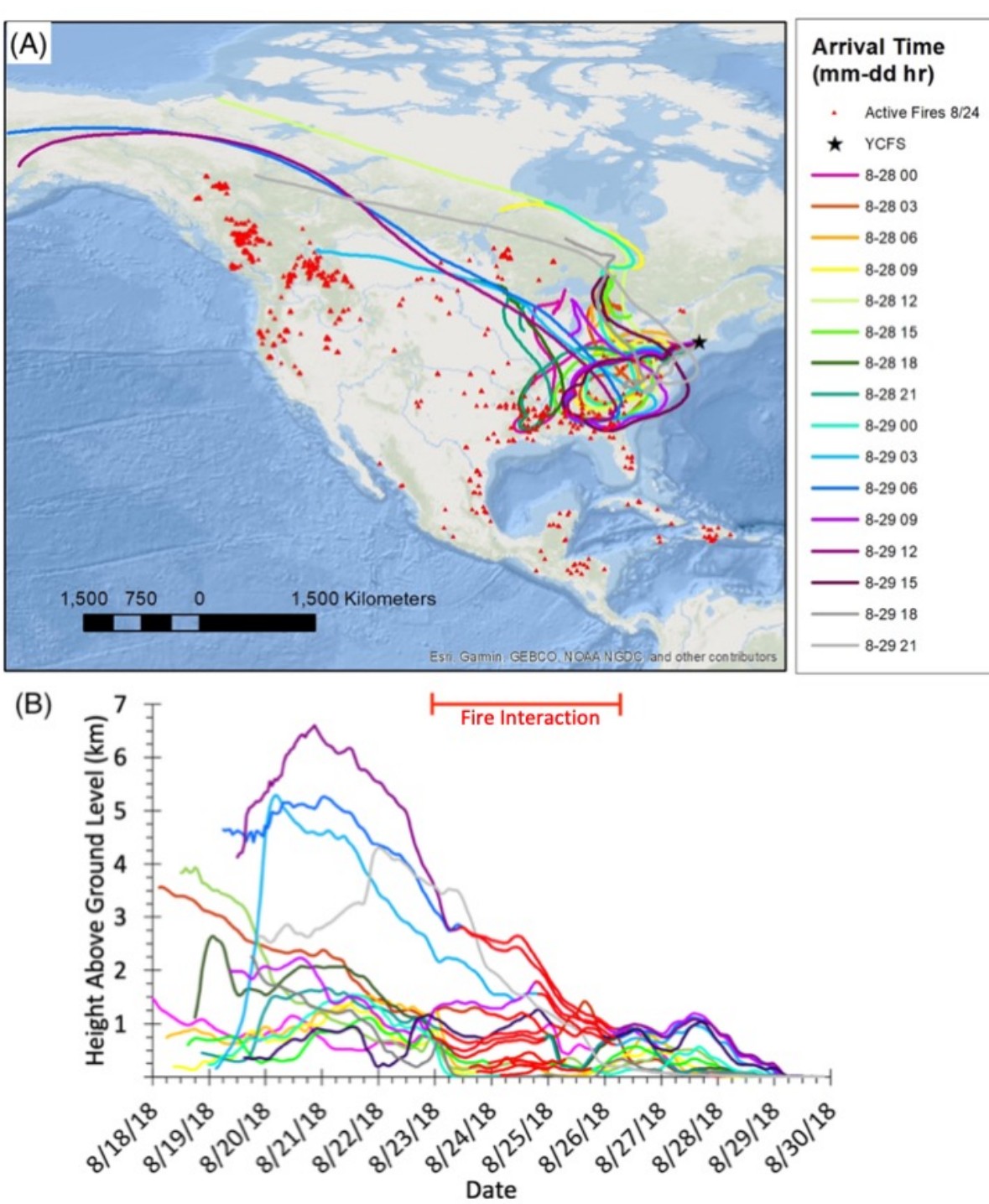

256

**Figure 6: NOAA HYSPLIT Backward-trajectory model results for air parcels arriving on August 28th and 29th, 2018 to surface-level YCFS site. Each color represents the backward-trajectory for an air parcel arriving every three hours throughout the course of the day. The location of fires on August 24th (when most trajectories intersect the fire zone in the southeast U.S.) is depicted with red triangles (from NOAA HMS fire maps). The top map (A) shows the full 10-day trajectory and the bottom graph (B) shows the vertical height of each air parcel along its trajectory as well as the time where it may have intercepted fire smoke plumes (highlighted in red on each individual trace). The last two trajectories on 8-29 have no major fire interaction, and have thus been colored grey.**

**4 Conclusions**

This study provides three pieces of evidence for the potential influence of long-distance transport of emissions from wildfires and other biomass burning on air quality in metropolitan NYC and the northeastern U.S. Together, surface-level measurements made at multiple regional sites, satellite smoke plume imagery, and air parcel backward-trajectory model results indicate that biomass burning smoke was transported to the metropolitan NYC area during two separate events in August leading to elevated levels of $PM_{2.5}$ and BC across the region. First, prolonged regional concentration enhancements in tracers associated with biomass burning–$PM_{2.5}$, BC, and CO–indicates the potential influence of biomass burning smoke on August 16th-17th and August 27th-29th. Second, NOAA Smoke Maps confirm the arrival and presence of smoke plumes over the Long Island Sound YCFS region on all five days of interest, and their absence after the events. Finally, backward-trajectory models provide additional information on the origin of air parcels and the associated pollutants. Air parcels from August 16th and 17th passed over western Canada, whereas air parcels arriving on August 28th and 29th passed over the southeastern U.S. The sets of trajectories during both events passed over regions with numerous active fires, including wildfires in western Canada, and most likely controlled agricultural burning in the southeast U.S. Regardless of the cause of the fire, these results show that fires in multiple places in North America can impact air quality in metropolitan NYC, in Connecticut, and more broadly, in the northeastern U.S.

This work, in conjunction with previous studies on the long-distance transport of biomass burning pollutants to other locations (Colarco et al., 2004; Cottle et al., 2014; Dreessen et al., 2016; Jung et al., 2016), reinforces the growing need to understand the long-range influence of wildfires. Increased understanding of long-distance transport is critical for predicting and managing air quality health risks in smoke-impacted areas. During both observed events (Figure 2), New York State issued air quality health advisories in New York City Metro and Long Island specifically for ozone (on August 16th, 28th, and 29th) (New York Department of Environmental Conservation, 2018), though the implications of the transported emissions for ozone production are not directly evaluated here. This long-distance transport process is also important since wildfire $PM_{2.5}$ has been specifically shown to have significant health effects with respiratory effects that possibly exceed those of other $PM_{2.5}$ sources, and multi-day wildfire smoke events have even been shown to have short-term health effects on susceptible populations (statistically-significant effects at concentrations >37 μg $m^{-3}$) (Liu et al., 2015;  Liu et al., 2017). As climate change continues to impact the likelihood, prevalence, and intensity

of wildfires across the U.S. and Canada, air quality scientists and policy makers must pay increasing attention to the
influence that these emissions have on air pollution issues, not only on a local scale but nationally and internationally.
This is critical as increased emissions throughout a prolonged fire season, when coupled with common meteorological
transport, can lead to enhanced background concentrations of primary $PM_{2.5}$ (including BC) and reactive precursors
to SOA and ozone (Akagi et al., 2011; Urbanski et al., 2008). In all, these two observed events in the NYC area in
August 2018 are examples that demonstrate the role of long-distance transport of biomass burning emissions as
important contributors to the evolving air quality challenges facing metro NYC and similar urban areas as local
emissions from controllable sources are further reduced (e.g. Khare & Gentner, 2018; NYC Dept. Health, 2018) and
wildfires become increasingly frequent.
**Author Contributions**
H.M.R., J.C.D., and D.R.G. designed the study and led analysis. J.C.D. managed instruments and collected data at the
YCFS. H.M.R. analysed data and compiled modeling results and satellite imagery. H.M.R. and D.R.G. wrote the paper
and all authors contributed to refining the manuscript.
**Competing Interests**
The authors declare that they have no conflicts of interest.
**Data availability**
Data are available upon request to the corresponding author and are in their respective public repositories (LISTOS
archive     [https://www-air.larc.nasa.gov/missions/listos/index.html],     EPA     Air     Quality     System
[https://www.epa.gov/aqs],     and     NOAA     Hazard     Mapping     System     Fire     and     Smoke     Products
[https://www.ospo.noaa.gov/Products/land/hms.html]).

**Acknowledgements**

We acknowledge the support of the Yale Department of Chemical and Environmental Engineering undergraduate research funding, the Yale Natural Lands Fund, the Peabody Museum, and the U.S. EPA. This publication was developed under Assistance Agreement No. RD835871 awarded by the U.S. Environmental Protection Agency to Yale University. It has not been formally reviewed by EPA. The views expressed in this document are solely those of the authors and do not necessarily reflect those of the Agency. EPA does not endorse any products or commercial services mentioned in this publication. Thank you as well to Lukas Valin (EPA), Pete Babich (CT DEEP), and David Wheeler and Dirk Felton (NYS DEC and SUNY-ASRC) for instrumentation at the YCFS and data from other sites including Queens and Pinnacle State Park; Rich Boardman, Tim White, and David Skelly (Yale/Peabody), and Ethan Weed and Amir Bond (Peabody EVOlutions Interns) for their help establishing the YCFS measurement site; and Paul Miller (NESCAUM) for organizing the LISTOS campaign. The authors gratefully acknowledge the NOAA Air Resources Laboratory (ARL) for the provision of the HYSPLIT transport and dispersion model used in this publication.

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
