# Peer review of "Evidence for impacts on surface-level air quality in the"

_Atmospheric Chemistry and Physics, 2019_

## Referee Comment (RC1) · Anonymous Referee #1 · 4 Sep 2019

This paper presents evidence for impacts on surface-level air quality, specifically PM2.5, BC, and CO, in the Northeastern U.S. from long-distance transport of smoke from North American fires in August 2018. They collected hourly data of PM2.5, BC, and CO concentration at the Yale Coastal Field Station (YCFS). In addition, they used publicly available monitoring data at five other locations. NOAA's smoke maps based on satellite imagery were used to provide information on the horizontal distribution and density of the smoke plumes across North America and the sampling region. The satellite imagery generally suggested that during the two fire episodes, large areas in

[Figure]

North America were affected by the smoke. Some inconsistencies between the satellite imagery and surface observation were explained as a result of unknown vertical distributions. In order to obtain insights on the origin of the surface air parcels, they further used NOAA HYSPLIT air parcel backward-trajectory models to provide additional information on the horizontally- and vertically-resolved transport pathways. They found that many of trajectories have intercepted locations with wildfire activities observed by satellite imagery. Air parcels in the first episode intercepted fire locations at 2-7 km above the ground level, whereas air parcels in the second episode were closer to the ground level which may also be affected by intentional crop fires in the southeastern U.S. They conclude that this work reinforces the growing need to understand the long-range influence of wildfires.

General comments:

I believe this work is technically sound and publishable, but I am not convinced that ACP is the right venue. Since the observation data is limited to PM2.5, BC, and CO, I must say that the contribution of this work in terms of providing new data beyond what is already available from routine monitoring is limited. Since the majority of the observational data (5 out of 6 sites), Smoke Maps, and back-trajectories are based on publicly available information, I believe there must be substantial merit in data analysis to warrant publication on ACP. However, it is not clear to me how observation of two events based on PM2.5, BC, and CO that may have originated from smoke plumes in the U.S. benefits the research community. Since the Smoke Maps showed nearly the entire U.S. was covered by smoke, it does not seem surprising that back-trajectories intercept with smoke plumes somewhere. I believe the manuscript should substantially expand on data analysis and demonstrate novelty to be considered for publication on ACP or should be published elsewhere.

Specific comments:

- It may be useful to contrast "Event" and "Non-event". If the same analysis is performed

on cleaner days between Event 1 and Event 2, do back-trajectories pass through any wildfire locations?

Typo - Line 83. Right parenthesis missing.

---

## Referee Comment (RC2) · Anonymous Referee #2 · 5 Sep 2019

Referee Comment ACP-2019-700

General Comment: The paper describes lines of evidence leading the authors to conclude that two pollution events experienced in the New York City Metro area and along coastal Connecticut during August 2018 were in large part attributable to emissions from biomass burning events. The paper is well written and nicely presented. There is nothing ground-breaking in the results, but it is a solid paper and deserves to be published largely as is.

[Figure]

Specific Comments: The authors are \*mostly\* good about being precise in their wording so as not to mislead the reader about what was actually observed. As someone who is sensitive to this I did find a few places where more precise wording is warranted. I have noted these instances as "Technical Corrections".

Lines 38-44: Missing in the introduction is any mention or discussion of aging and chemical transformations that occur in biomass burning plumes. For the present study the authors rely on "persistent" tracers that remain somewhat (or mostly) intact over the multiple days it takes to reach their measurement site. I'm not suggesting a detailed discussion here, but some acknowledgement of the process and how it might affect the study is needed. Maybe just a couple of sentences or a short paragraph?

Line 114: The very high CO spikes at the YCFS on 8/16 and 8/29 deserve some attention. It seems likely to me that these spikes are caused by "hyper-local" sources, and they are more than a factor of three greater than the high smoke influenced values and a factor of two higher than anything seen in Bridgeport (and Queens). Maybe a delivery truck idling near the inlet? Or a "dirty" ship sending a plume over the site? I suggest the authors look more carefully at their data to make sure these spikes are not caused by a local contamination source.

Lines 130-133: The authors should be aware (and potentially indicate in the paper) that August 5, 6, 7, 10, 16, 28, and 29 were all identified as "Air Quality Health Alert" days in New York State. In each case ozone was predicted to be the pollutant of greatest concern, but since high ozone and high PM2.5 often occur simultaneously, it is not surprising to have high PM levels on August 6, 7, and 10.

Line 250: Following up on this, the authors only mention that 8/29 was an air quality health alert day. The 16th and 28th (also study days) were also AQHA days for the NYC metro area or nearby communities.

Line 269: The data availability sentence seems a little terse. At least identify the public repositories.

[Figure]

Technical Corrections: Line 16: Insert "at surface-level sites" between "in" and "arriving".

Line 74: While the AE33 and 48i can be configured to provide 1 second data, I don't think that is the case for the BAM 1020. It is typically configured to provide only 1 hour averaged data.

Line 134: The identification of panels A and B are reversed.

Line 141: Change "in" to "over".

Line 146: Add "at YCFS" between "peaking" and "on".

Line 155: Suggest rewording this to read, "that smoke from aloft was available for transport to the surface, followed by the increases in . . .".

Lines 167 and 173: The words "compared to YCFS (starred)" are out of place in the first sentence. I suggest taking them out of this sentence, and adding a second sentence that simply reads, "The YCFS is indicated by a star."

Line 239: Change "in" to "over".

---

## Author Comment (AC1) · 10 Nov 2019

We greatly appreciate the reviewers' thoughtful comments and recommendations that have allowed us to improve our manuscript. After carefully reviewing their feedback, we have incorporated the following changes to our manuscript and provide the following responses.

Response to Referees' comments:

Anonymous Referee #1:

[Figure]

This paper presents evidence for impacts on surface-level air quality, specifically PM2.5, BC, and CO, in the Northeastern U.S. from long-distance transport of smoke from North American fires in August 2018. They collected hourly data of PM2.5, BC, and CO concentration at the Yale Coastal Field Station (YCFS). In addition, they used publicly available monitoring data at five other locations. NOAA's smoke maps based on satellite imagery were used to provide information on the horizontal distribution and density of the smoke plumes across North America and the sampling region. The satellite imagery generally suggested that during the two fire episodes, large areas in North America were affected by the smoke. Some inconsistencies between the satellite imagery and surface observation were explained as a result of unknown vertical distributions. In order to obtain insights on the origin of the surface air parcels, they further used NOAA HYSPLIT air parcel backward-trajectory models to provide additional information on the horizontally- and vertically-resolved transport pathways. They found that many of trajectories have intercepted locations with wildfire activities observed by satellite imagery. Air parcels in the first episode intercepted fire locations at 2-7 km above the ground level, whereas air parcels in the second episode were closer to the ground level which may also be affected by intentional crop fires in the southeastern U.S. They conclude that this work reinforces the growing need to understand the long-range influence of wildfires.

General Comments: I believe this work is technically sound and publishable, but I am not convinced that ACP is the right venue. Since the observation data is limited to PM2.5, BC, and CO, I must say that the contribution of this work in terms of providing new data beyond what is already available from routine monitoring is limited. Since the majority of the observational data (5 out of 6 sites), Smoke Maps, and back-trajectories are based on publicly available information, I believe there must be substantial merit in data analysis to warrant publication on ACP.

Response: We appreciate the reviewer's confidence in the quality of this work. In response to their concern, we affirm that the manuscript does generate new data and

contains a new synthesis of data across multiple platforms, which makes the paper more robust than the presentation of any of the sites, model, or satellite data products independently. The conclusions reached in this study are a product of this combined analysis, and provide science- and policy-relevant information that advance the research community's and Atmospheric Chemistry and Physics' objectives. Here, we specifically highlight the new data and results in this paper:

This paper's results specifically leverage completely new data from a new field site that our research group set up on the Coast of Connecticut—the Yale Coastal Field Station (YCFS). The YCFS is located strategically to minimize local urban Connecticut influences while also being in the greater Metro New York City area, making it an ideal location for such a study; the results here leverage its role as a regional background site (this clarification was added at lines 102-104). The other ambient data discussed here come from 5 other field stations, and are presented alongside our research group's data to demonstrate regional significance of the wildfire transport events and also their impacts on air quality in the New York City area. Analysis with this data has not been published elsewhere (to our knowledge), and the use of non-public data is not a requirement for publication in Atmospheric Chemistry and Physics. Additionally, our downwind measurements of black carbon represent a valuable contribution to the field; black carbon is not routinely monitored at many sites, as evidenced by the lack of any non-urban measurements in the greater NYC area. Without black carbon measurements, the role of forest fires in this combined dataset would be hard to assess.

The NOAA HYSPLIT model is publicly available, but we performed new model runs for the purposes of this manuscript. Similar to other studies (e.g. Cottle et al., 2014 in Atmospheric Environment; Diapouli et al., 2014 in Atmospheric Environment), we use the existing HYSPLIT model but generate new runs tailored to the conditions of interest in our study. Furthermore, the HYPSLIT model is used in a wide-ranging number of publications in the field.

Similarly, the NOAA Smoke Maps used here are publicly available, but we show them

in our manuscript as a complimentary data source that is independent of the other methods used. Previous studies have also relied upon smoke maps, such as Shrestha et al. (2019)'s paper "Impact of Outdoor Air Pollution on Indoor Air Quality in Low-Income Homes during Wildfire Seasons" (Int. J. Environ. Res. Public Health), to identify event and non-event days.

Comment: However, it is not clear to me how observation of two events based on PM2.5, BC, and CO that may have originated from smoke plumes in the U.S. benefits the research community.

Response: The urban air quality research community is working at a time when urban air quality is rapidly evolving with decreases in emissions from traditional (typically anthropogenic combustion-related) sources. This is increasing the relative impact of other sources that have either been under-regulated or are un-controllable (e.g. Khare & Gentner, 2018 ACP, Figure 2), for example, biomass burning. Cities like New York have also made extensive progress on reducing local emissions of PM2.5 and other pollutants (a great example can be seen in the New York City Community Air Quality Survey (NYCCAS), 2018). Thus the role of uncontrolled biomass burning emissions and their transport to urban areas like NYC is likely to become a larger fraction of PM2.5 contributions, which will be further exacerbated by increases in the frequency and magnitude of forest fires. For example, Figure 2 in our manuscript already shows major increases in PM2.5 above the baseline concentrations (~5 ug/m3; consistent with NYCCAS 2016 average range) during the biomass burning transport events (~20 ug/m3).

So, this manuscript benefits the research community by documenting the effects of cross-continental smoke transport on the New York City Metropolitan area. To our knowledge, no such study has been performed for the NYC metropolitan area, which is a megacity with a population of over 20 million. The findings are valuable for regional, national, as well as international air quality planning, forecasting, and management. In addition, understanding the impact of wildfire smoke on the NYC metropolitan area will

be critical for assessing human exposure to potentially hazardous components of wildfire smoke. These two events can serve as demonstrative examples to the research community that the long-distance transport of smoke from biomass burning has had an impact on the NYC metropolitan area. As wildfires become more frequent, it is valuable to have documented their consequences at national scales beyond the more-common regional scope, to foster future research and planning. To address the reviewer's question, we have added a sentence in the Conclusion section to further emphasize this point (Lines 345-349).

Comment: Since the Smoke Maps showed nearly the entire U.S. was covered by smoke, it does not seem surprising that back-trajectories intercept with smoke plumes somewhere.

Response: We acknowledge that in some (not all) of the smoke maps do show wildfire plumes aloft over several parts of the U.S., especially the Northern U.S., but the maps do reveal helpful spatial and temporal patterns in smoke coverage that supplement the other data and substantiate the conclusions. It was our intent for the smoke maps to provide different information and interpretation separate from the backward-trajectories. The smoke maps were used to demonstrate that smoke plumes were observed in the immediate vicinity of the YCFS and other nearby sites on the specific days when high concentrations of biomass burning-related tracer species were observed in metro NYC. The presence of a smoke plume above the YCFS confirmed that it was therefore possible that smoke had also had been transported to the surface. In this sense, the degree of smoke plume coverage across other locations in the US was not a determining factor in our assessment of whether or not the smoke maps confirmed the presence of smoke in the NYC metro area.

In contrast, the backward-trajectories were mapped in combination with the documented location of the fires before the observed increase in surface-level concentrations of PM2.5, BC, and CO at the YCFS. This approach was specifically used since some of the smoke maps indicate multiple, wide-ranging plumes in the U.S., and the

smoke maps did not provide any information on the vertical component of the smoke plumes. By illustrating that the backward-trajectories passed over the locations of the wildfires themselves, at altitudes where it was reasonable to expect the concentrated smoke plumes to rise to, we demonstrate that these air parcels could have feasibly picked up the concentrated pollutants associated with biomass burning.

To address the reviewer's concerns, a paragraph has been added (Lines 265-274) to clarify the difference in presentation and interpretation of the smoke maps and backward-trajectories. In addition, and in response to both this comment and the recommendation to analyze non-event days in addition to event-days (addressed below), we have added four examples of non-event days to the SI section (noted in main text Line 274). These examples show that on a day with low surface-level concentrations of PM2.5, BC, and CO (e.g. August 4th, 5th, and 21st), smoke maps do not show a plume above the NYC metropolitan region, nor do backward-trajectories have significant interaction with areas where active fires are burning. Also, we have clarified the language at lines 198-200 regarding the presence of aloft smoke plumes that could mix with the surface layer.

Comment: I believe the manuscript should substantially expand on data analysis and demonstrate novelty to be considered for publication on ACP or should be published elsewhere.

Response: We appreciate the referee's previous comments and suggestions, and refer to our longer response above related to novelty, justification, and merits of publication. In summary, our manuscript presents new data and a new multi-platform synthesis of 6 different ground sites, satellite observations, and pollutant transport modeling in order to conclude that emissions from biomass burning in 2 different North American regions were transported to metro NYC where they had a significant impact on regional PM2.5 concentrations. These results were more conclusive due to the multi-platform approach discussed here and due to the incorporation of data from a new strategically positioned field site—the YCFS. This study has scientific merit and policy implications

for air quality research, management, and planning in metro NYC (a non-attainment area). These findings will also be relevant to many other similar metropolitan areas, especially considering the increased propensity for wildfires with changing climate. Our manuscript shows the potential impact of forest fires, not just in the regions of the fires, but also in major urban areas on the other side of the continent. Based on all the reviewers' comments, we have made modifications and additions to improve the manuscript, including analysis of non-event days, clarification on our current analyses, and discussion of results.

Specific comments: Comment: It may be useful to contrast "Event" and "Non-event". If the same analysis is performed on cleaner days between Event 1 and Event 2, do backward-trajectories pass through any wildfire locations?

Response: Thank you for this valuable addition to our analysis. We have added a section in the SI to show parallel analysis for four examples of non-event days (Section S4). On August 4th, 5th, 13th, and 21st, surface level concentrations measured at the YCFS and other regional stations are lower than during the event days. In addition, NOAA Smoke Maps show no visible smoke clouds above the YCFS on August 4th, 5th, and 21st, and backward-trajectories have minimal interaction with active fires areas. August 13th shows some aloft plumes, but rain as well as over-water transport of air parcel back-trajectories before arriving at the YCFS would have reduced any potential surface-level contributions. While this analysis was not conducted for every single non-event day in the month of August, this provides several examples of non-event days in which the patterns observed in NOAA Smoke Maps and backward-trajectory models differ significantly from the patterns observed on event days.

We believe this additional analysis further strengthens the interpretation that during the two event days all three sources of data (field measurements, NOAA Smoke Maps, HYSPLIT backward trajectories) confirm a potential link to long-distance transport of smoke from biomass burning, which does not occur on days when surface-level concentrations are low.

Comment: Typo - Line 83. Right parenthesis missing.

Response: We have added the missing parenthesis.

—

Anonymous Referee #2:

General Comment: The paper describes lines of evidence leading the authors to conclude that two pollution events experienced in the New York City Metro area and along coastal Connecticut during August 2018 were in large part attributable to emissions from biomass burning events. The paper is well written and nicely presented. There is nothing ground-breaking in the results, but it is a solid paper and deserves to be published largely as is.

Response: We thank the reviewer for their support of our paper, and have addressed their specific comments below.

Comment: The authors are \*mostly\* good about being precise in their wording so as not to mislead the reader about what was actually observed. As someone who is sensitive to this I did find a few places where more precise wording is warranted. I have noted these instances as "Technical Corrections".

Response: We appreciate the technical recommendations for precise wording, which have been addressed in the technical corrections section, below.

Comment: Lines 38-44: Missing in the introduction is any mention or discussion of aging and chemical transformations that occur in biomass burning plumes. For the present study the authors rely on "persistent" tracers that remain somewhat (or mostly) intact over the multiple days it takes to reach their measurement site. I'm not suggesting a detailed discussion here, but some acknowledgement of the process and how it might affect the study is needed. Maybe just a couple of sentences or a short paragraph?

Response: Thank you for this recommendation. We have added some discussion

(lines 47-59) to acknowledge that aging and chemical transformation does occur in the smoke released from biomass burning. However, PM2.5, BC, and CO were selected because they have a relatively long atmospheric residence time compared to than other tracers. Although they become diluted during the long-distance transport, they are much less reactive than other chemicals released during biomass burning.

Comment: Line 114: The very high CO spikes at the YCFS on 8/16 and 8/29 deserve some attention. It seems likely to me that these spikes are caused by "hyper-local" sources, and they are more than a factor of three greater than the high smoke influenced values and a factor of two higher than anything seen in Bridgeport (and Queens). Maybe a delivery truck idling near the inlet? Or a "dirty" ship sending a plume over the site? I suggest the authors look more carefully at their data to make sure these spikes are not caused by a local contamination source.

Response: Thank you for the comment. We did not intend to infer that these spikes were related to the long-distance transport of smoke. We agree with the referee's interpretation that the extreme spikes that occurred at the YCFS on 8/16 and 8/29 are likely caused by a hyper-local source. We have added a sentence to the figure caption addressing these points as outliers and the potential that they were caused by a local source (lines 175, 181-183). However, the agreement in the baseline CO concentrations with the other sites reinforces the background CO concentrations, which is the primary purpose of the CO figure. Thus these 2 outlier spikes do not affect the broader interpretations of the data and resulting conclusions in the paper.

Comment: Lines 130-133: The authors should be aware (and potentially indicate in the paper) that August 5, 6, 7, 10, 16, 28, and 29 were all identified as "Air Quality Health Alert" days in New York State. In each case ozone was predicted to be the pollutant of greatest concern, but since high ozone and high PM2.5 often occur simultaneously, it is not surprising to have high PM levels on August 6, 7, and 10.

Response: We thank the referee for bringing these other dates to our attention. We

have added a sentence at the end of the paragraph (lines 163-165) acknowledging that these days were health advisory days. We fully agree that elevated PM2.5 often occurs with ozone due to the secondary production of aerosols. Because we have BC measurements to accompany the PM2.5 data, we are confident to ascribe "Events 1 and 2" (labeled in Figure 2) to biomass burning transport.

Comment: Line 250: Following up on this, the authors only mention that 8/29 was an air quality health alert day. The 16th and 28th (also study days) were also AQHA days for the NYC metro area or nearby communities.

Response: We have modified this sentence to include reference to all three advisory days which occurred during the two identified events (lines 325-326). Thank you for providing this additional information.

Comment: Line 269: The data availability sentence seems a little terse. At least identify the public repositories.

Response: We have added the names of the public repositories to modify the tone of this sentence.

Technical Corrections: Response: We appreciate the referee's careful review of the manuscript's language and technical suggestions. Our responses grouped together were appropriate.

Comment: Line 16: Insert "at surface-level sites" between "in" and "arriving".

Response: The wording has been modified.

Comment: Line 74: While the AE33 and 48i can be configured to provide 1 second data, I don't think that is the case for the BAM 1020. It is typically configured to provide only 1 hour averaged data.

Response: This sentence has been modified to reflect the correct time interval of the BAM 1020 (lines 91-92).

Typographical Comments: Line 134: The identification of panels A and B are reversed. Line 141: Change "in" to "over". Line 146: Add "at YCFS" between "peaking" and "on". Line 155: Suggest rewording this to read, "that smoke from aloft was available for transport to the surface, followed by the increases in . . .". Lines 167 and 173: The words "compared to YCFS (starred)" are out of place in the first sentence. I suggest taking them out of this sentence, and adding a second sentence that simply reads, "The YCFS is indicated by a star." Line 239: Change "in" to "over".

Response: Thank you. All of the above technical language issues have been corrected.